# The Provenance of Slovenian Milk Using ^87^Sr/^86^Sr Isotope Ratios

**DOI:** 10.3390/foods10081729

**Published:** 2021-07-27

**Authors:** Staša Hamzić Gregorčič, Nives Ogrinc, Russell Frew, Marijan Nečemer, Lidija Strojnik, Tea Zuliani

**Affiliations:** 1Department of Environmental Sciences, Jožef Stefan Institute, Jamova 39, 1000 Ljubljana, Slovenia; stasa.gregorcic@ijs.si (S.H.G.); nives.ogrinc@ijs.si (N.O.); lidija.strojnik@ijs.si (L.S.); 2Jožef Stefan International Postgraduate School, Jamova 39, 1000 Ljubljana, Slovenia; marijan.necemer@ijs.si; 3Department of Chemistry, University of Otago, P.O. Box 56, Dunedin 9016, New Zealand; russell.frew@otago.ac.nz

**Keywords:** geographical origin, milk, cow diet, ^87^Sr/^86^Sr, discriminant analysis

## Abstract

This work presents the first use of Sr isotope ratios for determining the provenance of bovine milk from different regions of Slovenia. The analytical protocol for the determination of ^87^Sr/^86^Sr isotope ratio was optimised and applied to authentic milk samples. Considerable variability of ^87^Sr/^86^Sr ratios found in Slovenian milk reflects the substantial heterogeneity of the geological background of its origin. The results, although promising, cannot discount possible inter-annual or annual variation of the Sr isotopic composition of milk. The ^87^Sr/^86^Sr ratios of groundwater and surface waters are in good correlation with milk, indicating that the Sr isotopic fingerprint in milk is reflective of cow drinking water. The ^87^Sr/^86^Sr ratio has the potential to distinguish between different milk production areas as long as these areas are characterised by geo-lithology. Discriminant analysis (DA) incorporating the elemental composition and stable isotopes of light elements showed that ^87^Sr/^86^Sr ratio together with *δ*^13^C_cas_ and *δ*^15^N_cas_ values have the main discrimination power to distinguish the Quaternary group (group 6) from the others. Group 1 (Cretaceous: Carbonate Rocks and Flysch) is associated with Br content, 1/Sr and *δ*^18^O_w_ values. The overall prediction ability was found to be 63.5%. Pairwise comparisons using OPLS-DA confirmed that diet and geologic parameters are important for the separation.

## 1. Introduction

Proof of provenance has increased in relevance over the past decade because of its positive impact on food safety, quality and consumer protection per national legislation and international standards and guidelines. This trend also coincides with an increase in consumer demand for local and regional food, which is considered higher quality, safer and more sustainable. This has created interest in building local and regional food systems across Europe, including Slovenia. Most milk and dairy products, produced and processed in Slovenia, now use the “Selected Quality—Slovenia” mark, which indicates that the product is of Slovenian origin. The characteristics of milk are highly dependent on the farming practices and the soil where cattle graze, thereby, from the geographical region, reflecting specific and peculiar geologic information. Consequently, the geographic origin of milk and dairy products is an important factor affecting quality.

The geographical origin of milk and especially dairy products has been frequently traced by using stable isotope analysis of light elements (*δ*^2^H, *δ*^13^C, *δ*^15^N, *δ*^18^O and *δ*^34^S) [1,2,3,4] or in combination with the multi-elemental analysis [5,6,7,8]. Recently, isotopic information from heavy elements in soil and food has been explored for its potential to serve as reliable geographical tracers for food origin. In particular, the isotopic composition of strontium (Sr) has proven to be a promising tool for discriminating at a regional level. It has been found that the Sr retains its original isotopic ratio unaltered up to the end product, even after processing (e.g., mixed biological processes involved in soil-plant interaction), with no isotopic fractionation [9]. The ^87^Sr/^86^Sr isotope ratio, therefore, provides a unique and highly efficient geographical tracer for several types of food products such as asparagus [10], rice [11,12], tea leaves [13,14], coffee [15,16], orange juice [17], and wine [18,19,20]. The authors of these studies highlighted two main applications of the ^87^Sr/^86^Sr isotope ratio: (i) to characterise the Sr isotope composition of a specific agricultural area, creating a valuable database for subsequently verify products cultivated in that area; and (ii) to discriminate with certainty different production areas of a specific product.

Although Sr isotopic analysis has led to significant advances in food traceability, there is still a lack of standard protocols for sample preparation, preservation and analysis, making data interpretation and cross-study comparisons difficult and confusing. The ^87^Sr/^86^Sr isotopic analysis in complex food matrices often requires extensive sample pretreatment and/or isolation steps before instrumental analysis, especially when multi-collector inductively coupled plasma mass spectrometry (MC ICP-MS) and thermal ionisation mass spectrometry (TIMS) are used [21,22]. The ^87^Sr/^86^Sr isotope ratio determination is influenced by the isobaric interference of ^87^Rb, leading to incorrect isotope ratio determination [23]. Among four stable isotopes of Sr (^84^Sr, ^86^Sr, ^87^Sr, and ^88^Sr), only ^87^Sr is radiogenic and formed by *ß*-decay of ^87^Rb. Thus, ^87^Rb and ^87^Sr are often co-present in environmental samples. Chemical separation of ^87^Rb from ^87^Sr is required to assure accurate results and is usually accomplished using separation techniques such as extraction chromatography. Among the extraction chromatographic resins developed for the isolation of Sr from the complex sample matrix, Sr resin seems to be the most popular one due to its high selectivity for Sr [10,12,17,24]. The extractant contains 4,4′(5′)-di-t-butylcyclohexano 18-crown-6 (crown ether) in 1-octanol on inert polymeric support with the working capacity of 8 mg Sr per 2 mL column. As the column and resin size are dependent on the concentration of the elements present in the sample matrix under study, the efficiency of Sr-matrix separation also depends on the volume and molarity of nitric acid (HNO_3_). Horwitz et al. (1992) [25] show that the Sr capacity factor increases with increasing HNO_3_ concentration and decreases with increasing concentration of other cations such as barium.

Milk is a complex matrix with a higher organic load (fat and proteins) than most other food products previously cited, making the determination of Sr isotope ratios challenging. Although there are studies on the ^87^Sr/^86^Sr isotope ratio in dairy products [24,26,27,28], to our knowledge, only a limited number of studies on the ^87^Sr/^86^Sr isotope ratio determination in milk has been published [29,30]. Since no standard separation procedure for milk exists, there was a need for a methodology for accurately measuring Sr isotope ratios. To overcome the lack of standard protocols for sample preparation and analysis needed to enable comparability of cross studies and data interpretation, a proposed method for the ^87^Sr/^86^Sr isotope ratio determination in milk was optimised within this work. The method was shown to be fit-for-purpose for determining milk provenance and could be used in real-world applications. 

Therefore, our study aimed to determine the ^87^Sr/^86^Sr isotope ratio in milk using an optimised and validated analytical method and apply it to Slovenian milk samples originating from different geological regions to test its applicability in traceability studies.

## 2. Materials and Methods

### 2.1. Sampling

Milk samples were collected from 43 dairy farms located in different geological areas in Slovenia in 2014 (Figure 1).

The climatic conditions in Slovenia do not allow year-round grazing on outdoor pastures. Additionally, the landscape is diverse, where not all areas allow the growing of appropriate feed, so the geographical origin of winter feed may change. Both circumstances are responsible for the change in the cow’s diet. To evaluate these changes, milk samples were sampled during summer and winter in 2014 and the winter of 2015. Further, the elemental and stable isotopic composition of light elements (H, C, N, O and S) of authentic milk samples was determined to characterise authentic Slovenian milk. 

### 2.2. ^87^Sr/^86^Sr Isotope Ratio in Authentic Slovenian Milk

Although milk contains about 87% of water, its proteins, carbohydrates, and especially fat make its matrix very complex in the Sr isotope analysis. If not adequately destroyed, organic remnants can irreversibly adsorb on the extraction resin, thus reducing its exchange capacity and leading to a reduction in the Sr recovery and possibly to isotopic fractionation. The method was optimised in terms of completeness of mineralisation, chemical recovery of Sr isolated from the sample matrix (Appendix A), minimal contamination, and turnaround time. The blanks of analyte-free media were prepared using the same materials and reagents as for the samples. A procedure for optimisation and validation of the analytical method for accurate ^87^Sr/^86^Sr isotope ratio measurement in milk is fully described in Appendix A, while the analytical protocol is presented in Figure 2. 

In summary, as presented in Figure 2, for pretreatment of the milk samples, 0.30 g were subjected to microwave digestion and then evaporated to dryness and redissolved in 1 mL of 8M HNO_3_. A column (2 mL) was filled with 0.30 g resin, activated by washing with HCl. The resin was acidified with 3 mL HNO_3_ before sample loading to prevent any loss of Sr. Subsequently, the sample solution was loaded onto the column. Rb was eluted with 5 mL of 8M HNO_3_, after which the Sr was collected in purified water washes (Appendix A). The Sr solution was then evaporated and purified again through extraction separation. Finally, the ^87^Sr/^86^Sr isotope ratios were determined using MC ICP-MS.

#### Isotope Analysis Using MC ICP-MS

Strontium isotope ratio determinations were carried out using a Nu II multi-collector ICP-MS instrument (Nu Instruments, Ametek Inc., Wrexham, UK) fitted to an Aridus II^TM^ Desolvating Nebulizer System (Teledyne Cetac, Omaha, NE, USA) by the procedure of Zuliani et al. (2020) [32]. All samples were run in a standard-sample-standard bracketing sequence using standard Sr isotopic solution (NIST SRM 987: *Strontium carbonate*; ^87^Sr/^86^Sr_certfied_ = 0.71034 ± 0.00026; National Institute of Standards and Technology, Gaithersburg, MD, USA).

### 2.3. Multi-Elemental Analysis Using EDXRF

The multi-elemental composition of milk, including Sr stable isotope ratio, was performed using freeze-dried and homogenised milk samples. Energy-dispersive X-ray fluorescence spectrometry was used to determine the following elements: calcium (Ca), chloride (Cl), potassium (K), phosphorus (P), sulphur (S), bromide (Br), rubidium (Rb), and strontium (Sr). Each milk sample (0.5–1.0 g) was pressed into a pellet using a hydraulic press. As primary excitation sources, the annular radioisotope excitation sources of Fe-55 (10 mCi) and Cd-109 (20 mCi) from Isotope Products Laboratories (Valencia, CA, USA) were used. The emitted fluorescence radiation was measured using an energy dispersive X-ray spectrometer composed of a Si(Li) detector (Canberra Industries, Meriden, CT, USA), a spectroscopy amplifier (M2024, Canberra Industries, Meriden, CT, USA), ADC (M8075, Canberra Industries, Meriden, CT, USA) and PC based MCA (S-100, Canberra Industries, Meriden, CT, USA). The spectrometer was equipped with a vacuum chamber. The energy resolution of the spectrometer was 175 eV at 5.9 keV. An analysis of the X-ray spectra was made using the AXIL (IAEA, Vienna, Austria) spectral analysis program [33,34].

Sample preparation and the analytical procedure were critically tested and evaluated according to uncertainty, accuracy, and limits of detection (LOD) in our previous investigation [35].

### 2.4. Isotope Ratio Mass Spectrometry (IRMS) Measurements

Stable isotope ratio measurements were performed using isotope ratio mass spectrometry (IRMS) and reported using the *δ*-notation in ‰ using Equation (1) [36]:(1)δ(i/jE)=δi/jE=i/jRP−i/jRRefi/jRRef
where superscripts *i* and *j* denote the highest and the lowest atomic mass number of element *E*, and *R_P_* and *R_Ref_* indicate the ratio between the heavier and the lighter isotope (^2^H/^1^H, ^13^C/^12^C, ^18^O/^16^O, ^15^N/^14^N, ^34^S/^32^S) in the sample (*P*-product) and reference material (*Ref*), respectively. The *δ*^2^H and *δ*^18^O values are reported relative to the V-SMOW (Vienna-Standard Mean Ocean Water) standard, *δ*^13^C values to the V-PDB (Vienna-Pee Dee Belemnite) standard, and the *δ*^34^S sulphur values relative to the V-CDT (Vienna Cañon Diablo Troilite) standard. The *δ*^15^N values are reported relative to AIR. 

The ^18^O/^16^O ratio in milk water (*δ*^18^O_w_) was determined directly in milk using the equilibration method where the sample was purged with a reference CO_2_/He gas (5% CO_2_, 95% of He) at 40 °C for three hours. Measurements were performed using a Multiflow system (IsoPrime, Cheadle Hulme, Manchester, UK) connected to a continuous flow IRMS (GV Instruments, Manchester, UK). Analyses were calibrated against two internal laboratory reference materials: Snow water (*δ*^18^O = −19.73 ± 0.02‰) and seawater (*δ*^18^O = −0.34 ± 0.02‰). For independent control, laboratory reference material Milli-Q water was used as control material (*δ*^18^O = −9.12 ± 0.04‰). The internal laboratory and independent laboratory reference materials were calibrated against international reference materials: V-SLAP2 (Standard Light Antarctic Precipitation, *δ*^18^O = −55.5 ± 0.02‰) and V-SMOW (Vienna-Standard Mean Ocean Water 2, *δ*^18^O = 0 ± 0.02 ‰).

Further, ^13^C/^12^C, ^15^N/^14^N and ^34^S/^32^S ratios were determined in casein samples. Milk fat was removed by centrifugation (Type Centric 322 A, TEHTNICA, Železniki, Slovenia, 10 min at 3200 rpm), and casein by precipitation from the skimmed milk by acidification at pH 4.3 with 2M HCl (Carlo Erba, Val de Reuil, Italy) followed by centrifugation for 10 min at 3200 rpm. The precipitate was rinsed twice with Milli-Q water (Millipore, Burlington, MA, USA), followed by acetone and petroleum ether (Carlo Erba, Val de Reuil, Italy) and freeze-dried [37].

The freeze-dried casein sample was transferred to a tin capsule, closed with tweezers and placed into the autosampler of the elemental analyser. For ^13^C/^12^C, ^15^N/^14^N and ^34^S/^32^S determination, 10 mg of casein samples were analysed simultaneously using the IsoPrime 100-Vario PYRO Cube (OH/CNS Pyrolyser/Elemental Analyser) (IsoPrime, Cheadle, Hulme, UK). The results were calibrated against the international standards: IAEA-CH-7 (*δ*^13^C = −32.15 ± 0.03‰), IAEA-CH-6 (*δ*^13^C = −10.45 ± 0.03‰), IAEA-CH-3 (*δ*^13^C = −24.72 ± 0.04‰), IAEA-S-1 (*δ*^34^S = −0.3‰), IAEA-S-2 (*δ*^34^S = +22.49 ± 0.16‰). Other reference materials included: CRP-IAEA casein (*δ*^13^C = −20.3 ± 0.09‰, *δ*^15^N = +5.62 ± 0.19‰, *δ*^34^S = +4.18 ± 0.74‰), and casein, B2155 Sercon (*δ*^13^C = −26.98 ± 0.13‰, *δ*^15^N = +5.94 ± 0.08‰, *δ*^34^S = +6.32 ± 0.8‰).

### 2.5. Statistical Analysis

All samples were prepared in triplicate, and the data are presented as mean with standard deviation (SD) of triplicate independent experiments. Statistical analysis was performed using the XLSTAT software package (Addinsoft, New York, NY, USA). Simple statistical analyses were carried out, including an analysis of variance (ANOVA) with the Mann–Whitney (MW) and Kruskal–Wallis (KW) tests, since the data are not normally distributed. Furthermore, to determine the key factors responsible for differentiation of the region of the geographical origin of milk, a discriminant analysis (DA) was used. Moreover, orthogonal partial least squares discriminant analysis (OPLS-DA) was introduced for pairwise comparisons among two overlapping geological groups using the SIMCA^®^ software package (Umetrics, Umea, Sweden).

## 3. Results and Discussion

### 3.1. Strontium Isotope Ratio of Authentic Slovenian Milk

The first values for the ^87^Sr/^86^Sr ratio in Slovenian milk samples (*n* = 77) are presented in Table 1. Slovenia is a relatively small country covering a mere 20,273 km^2^ but boasts great diversity in complex geology, relief, hydrological systems, and vegetation. Unfortunately, this diversity was not observed to the same extent in the analysed milk samples.

The ^87^Sr/^86^Sr ratios in the milk samples collected from farms at different locations showed a moderate degree of variation, spanning from 0.708 to 0.713. When comparing the ^87^Sr/^86^Sr ratios between samples collected during summer and winter seasons of 2014 (Table 1), a certain degree of variability for some samples was observed; however, the differences were not statistically significant (Mann–Whitney; *p* = 0.9623). Further, no statistical difference was observed in ^87^Sr/^86^Sr ratios according to the year of production (Mann–Whitney; *p* = 0.1318). The same conclusion may be drawn from the concentrations of Sr in the milk samples. On the other hand, the reported δ^13^C and δ^15^N data of Slovenian milk reflect intra-annual changes in diet [8].

The Kruskal–Wallis test indicates that only four parameters are significantly related to the geological region (*p* < 0.001): ^87^Sr/^86^Sr ratios, *δ*^13^C_cas_, *δ*^15^N_cas_ and Br.

The relationship between ^87^Sr/^86^Sr ratios in the milk samples and rock type at each sampling location was also explored. The type and age of the soil were obtained from the geological map provided by the Geological Survey of Slovenia [31] (Figure 1). The ^87^Sr/^86^Sr isotope ratios in milk samples studied are in line with the isotopic values predicted for Slovenia, according to Hoogewerff et al. (2019) [38]. By their model, the soil ^87^Sr/^86^Sr ratios of most of Slovenia’s central and western parts should be in the range of 0.708 to 0.709. The ^87^Sr/^86^Sr ratios should be higher in the north-eastern part, ranging from 0.710 to 0.712. The values found in milk in the present study are in agreement with the modelled values.

Moreover, this information is in line with the bedrock composition and age. Indeed, most of the Slovenian territory is covered by tertiary and quaternary dolomites, limestones and alluvial deposits such as sandstones and claystones. There is a slight difference between milk samples from locations with quaternary alluvial deposits with alumo-silicate rocks with ^87^Sr/^86^Sr ratios ranging between 0.710 and 0.712, and locations with limestone and dolomite bedrock with ^87^Sr/^86^Sr ratios in the range from 0.708 to 0.710. On closer examination of the regional Slovenian milk samples, the overlap highlights the similarity between the geological and pedological characteristics of originating regions (Figure 3).

The data were compared with the Slovenian truffles, which have ^87^Sr/^86^Sr ratios ranging from 0.710 to 0.713 [39]. The values correspond to Slovenian milk samples except for the highest ^87^Sr/^86^Sr ratio of 0.71375 determined in truffles from Bloke, a karst plateau. When comparing the Sr isotopic ratios in dairy products originated from other countries with Slovenian milk, the span of the ^87^Sr/^86^Sr expressed in lower values has been recorded for cheese from Germany and Switzerland [24] and New Zealand [29].

In contrast, the ^87^Sr/^86^Sr ratios in milk and cheese from Quebec vary with a wide range of values, from 0.70961 up to a maximum of 0.71447, indicating a relative enrichment with radiogenic isotope ^87^Sr in Proterozoic and during the Paleozoic carbonate intrusive and limestone rocks composing the St. Lawrence Platform [30,40]. The large variability of Sr ratios in dairy products reflects the vast diversity of underlying bedrock and soils formed from them. Therefore, the widely scalable results of Sr ratios reflect the substantial heterogeneity of the geological background of its origin.

A specific pattern among samples was observed when comparing the Sr isotopic and elemental signatures in Slovenian milk samples based on geology (Figure 4).

Although several samples overlap, two trends can be identified: the first with high Sr concentration and high ^87^Sr/^86^Sr ratios (>0.7110) mainly from areas with quaternary alluvial deposits with alumo-silicate rocks and the second one related to lower ^87^Sr/^86^Sr ratios (<0.7090) at carbonate dominated areas. The overlapping values can be explained by: (i) different weathering rates of specific minerals in the rocks and soils, movement of water and sediments in a grazing area can influence Sr and Rb contents in milk samples leading potentially to different ^87^Sr/^86^Sr isotope ratios [27,41], (ii) the consumption of imported plants, particularly those enriched with high Ca and Sr content, can significantly alter the ^87^Sr/^86^Sr signatures in dairy products, even when consumed in small amounts. Thus, a consideration of total dietary intake is necessary when interpreting ^87^Sr/^86^Sr results.

The second source of Sr in milk is related to the drinking water supply. In Slovenia, most of the drinking water originates from groundwater and especially in the karst regions of the Sava River watershed, where river water represents the primary source of groundwater [42,43]. Therefore, we compared the ^87^Sr/^86^Sr ratios of milk with unpublished data of ^87^Sr/^86^Sr in the Sava, Ljubljanica, Pivka, Kamniška Bistrica and Logaščica rivers and rivulets and those determined in some mineral and spring bottled waters [32]. For comparison, we selected locations that lie close to the rivers for which the ^87^Sr/^86^Sr ratios are available. A good correlation between milk and groundwater data was observed (Figure 5), indicating that groundwater can represent an important source of Sr. 

However, it is interesting to note that most of the ^87^Sr/^86^Sr values for milk are higher than their corresponding river samples. One of the possible explanations could be the use of agricultural lime for soil improvement in fertile areas present mainly in the eastern part of Slovenia. This part is also known for its intensive agricultural practices where some field areas in specific locations are used to produce fodder plants for feeding livestock. It has been reported that the application of agricultural lime to low-calcareous soils can significantly lower the ^87^Sr/^86^Sr ratio of the watershed [44]. In these areas, maise silage is detected in more than 80% of the milk samples. 

Given that the cow’s body is up to 70% of water, the ^87^Sr/^86^Sr analysis of local drinking water might be helpful. Livestock in the Pannonian region is fed on the locally produced food, which also confirms the result of milk from Radenci (^87^Sr/^86^Sr = 0.71119), matching the ^87^Sr/^86^Sr ratio of the mineral water from the source Radenci (0.71120). This finding aligns with the investigation performed in the Parmigiano Reggiano milk and cheese production area [45]. In her study, the ^87^Sr/^86^Sr isotope ratio on water, whole milk, and diet samples allowed the construction of a linear relationship with multiple independent variables, from which the ^87^Sr/^86^Sr ratio of the milk is mainly correlated with the ^87^Sr/^86^Sr ratio of the hay. Thus, results indicate milk samples reflect the ^87^Sr/^86^Sr ratio of the feed linked to the soil and water. This is also in agreement with Stevenson et al. (2015) [30], in which the authors demonstrated a good correlation between the Sr isotopic composition of milk, cheese, and the bedrock geology of the dairy farm locations.

### 3.2. Discriminant Analysis

In the next step, we check if the ^87^Sr/^86^Sr ratio can increase the differentiation of Slovenian milk samples according to the geological region using different statistical approaches. In our statistical evaluation, stable isotope and elemental composition in milk samples were also included. The data are presented in Appendix A, while the detailed description of these parameters according to geographical origin is described in Potočnik et al. [8].

Sixty-three milk samples of four different geological regions (1—Cretaceous: Carbonate Rocks and Flysch, *n* = 8; 2—Jurassic-Triassic: Carbonate Rocks, n = 15; 3—Neogene: Carbonate Rocks, Paleogene: Deposits, *n* = 17; 6—Quaternary: Deposits, *n* = 23) and twenty-two parameters including ^87^Sr/^86^Sr, *δ*^18^O_w_, *δ*^13^C_cas_, *δ*^15^N_cas_, *δ*^34^S_cas_, Mn, Fe, Cu, Rb, Sr, Ca, K, Cl, S, P, Zn, Br, 1/Sr, Rb/Sr, Ca/Sr and K/Rb were processed by DA. In Figure 6, DA modelling results were shown as a discriminant function score plot (a) and a discriminant loadings plot (b). 

In the functional score plot, each group (centroid) is represented by a scatter plot, while in the loadings plot, they appear as a set of vectors indicating the degree of association of the corresponding initial variables with the first two discriminant functions. In the latter, the degree of distribution of each parameter in the classes is revealed. The first two discriminant functions accumulated 89.2% of the total variability. Two groups (groups 1 and 6) show a good tendency of separation among each other and from groups 2 and 3, which overlap slightly. Group 1 (Cretaceous: Carbonate Rocks and Flysch) is positioned in the right part of DA graph and is associated with the vectors of Br, 1/Sr and *δ*^18^O_w_. The mean values of these parameters in the centroid are the highest and the most influential for the separation. Inspection of the mentioned parameters with KW test reveals that they are significant for separating group 1 from the rest. A substantial amount of Br indicates that geologically is associated with a marine basement rocks origin. Higher *δ*^18^O_w_ are also typical for coastal regions. Further, group 6 positioned in the upper right part of the plot a is associated with vectors ^87^Sr/^86^Sr, *δ*^13^C_cas_ and *δ*^15^N_cas_ and according to KW test significant for discrimination among groups 1, 3 and 6. This group is located in the eastern part of Slovenia, located in Quaternary deposits, and it is also related to intensive milk production with higher content of corn in cow feed. Group 3, located in the lower right part of biplot a, is associated with Sr vectors, and inspection by KW and ANOVA tests reveal that both are significant for separation. Groups 2 and 3 are located in the lower part of biplot a, and here, *δ*^15^N_cas_ and Br are significant for discrimination between both groups. The prediction ability was the highest for the Quaternary group (91.3%) and the lowest for group 3 (Neogene + Paleogene; 41.2%), with an overall prediction of 63.5%.

Further, OPLS-DA tests for pairwise comparisons among two overlapping geological groups (2—Jurassic + Triassic, 3—Neogene + Paleogene: Figure 7) was calculated similarly to in the study performed by Chung et al. (2020) [46]. This model had an explanatory power of 94% (F1) for variation in the X variables and displayed high quality, goodness of fit, and predictability. It was found that the separation of these two groups is governed by *δ*^15^N_cas_ values govern, concentrations of Br and Rb/Sr ratio, indicating that not only geologic parameters are important for the separation, but also the way of cow feed and milk production—intensive with more corn silage or grass silage representative of the Jurassic + Triassic group.

## 4. Conclusions

In this study, we investigated the feasibility of the Sr isotope ratio analysis, combined with multivariate statistical analysis to discriminate milk samples from Slovenia based on their provenance. The ^87^Sr/^86^Sr ratios in milk samples were determined using an optimised method, which showed sufficient precision and accuracy to detect variations in Sr isotopic compositions between milk samples. Although Slovenia covers a relatively small area, its geology, geography and climate vary substantially. Large regional variability of ^87^Sr/^86^Sr ratios in Slovenian milk was observed, overlapping with other regions’ values. Thus, a complete separation of the regions based solely on the ^87^Sr/^86^Sr ratio of the milk was not possible. However, it was found that a combination of Sr isotopic profiling coupled to multivariate analysis is a promising tool for characterising milk according to geological origin. The milk produced in the Quaternary areas had high Sr content and higher ^87^Sr/^86^Sr values and differed from those produced in carbonate dominated areas with lower ^87^Sr/^86^Sr values.

In conclusion, the ^87^Sr/^86^Sr ratio can distinguish between different dairy areas as long as geolithological differences characterise these areas. In cases of a similar geological environment, combining elemental concentrations and isotope ratios, both light and heavy isotopes, might be advantageous. However, this approach is limited in the case of Slovenian milk. The close distance between macro-regions in Slovenia and the variations in climate affecting these regions make discrimination between milk samples of different origins more difficult, particularly when milk samples originate from locations positioned close to a zone between two or more regions and thus share a similar isotopic signature. 

Further, it has been confirmed that the cow’s diet and geologic parameters are important for the separation. Indeed, our study shows the correlation between the isotope ratio of strontium in milk and possible source of drinking water, in which diverse sources of strontium from the environment are reflected. However, to better understand the influence of different factors, i.e., water, feed and supplements, on the Sr isotope ratio in the milk samples, future research should investigate the ^87^Sr/^86^Sr ratio utilising paired samples of feed, water, and soil originating from the same location as the milk. In the perspective of food traceability, the database of the ^87^Sr/^86^Sr values in soils and waters in Slovenia could be also beneficial for future studies of local foods, where it can be used as a reference map to identify the authenticity of particular food product, or whether there are any unexpected isotopic variations.

## Figures and Tables

**Figure 1 foods-10-01729-f001:**
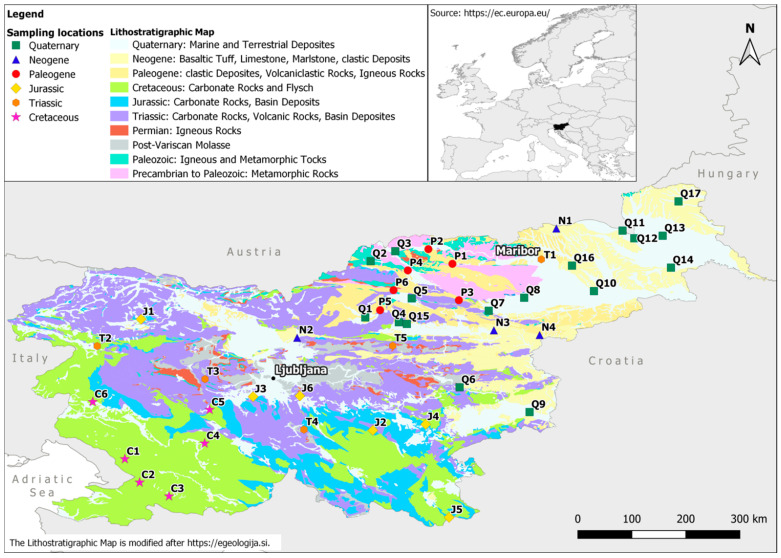
Geological map of Slovenia as indicated [31] with dairy farm locations (Table 1). The map was prepared by J. Vrzel.

**Figure 2 foods-10-01729-f002:**
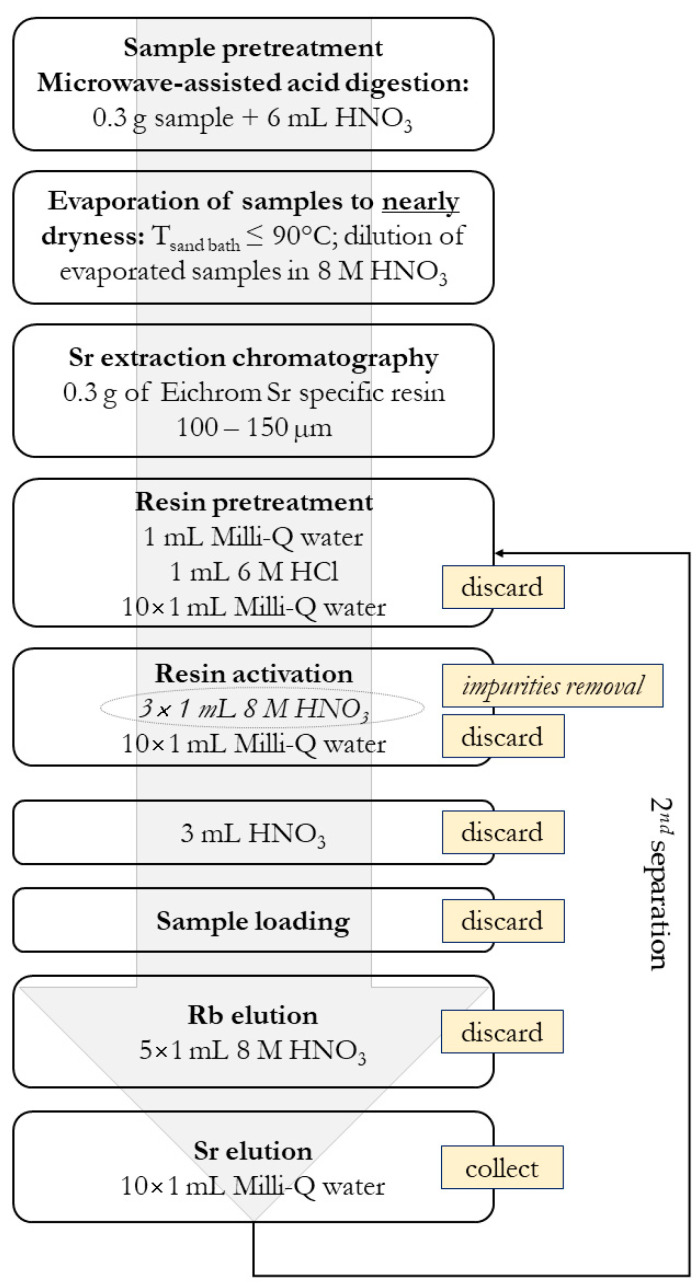
Analytical protocol for effective Sr isolation from the sample matrix.

**Figure 3 foods-10-01729-f003:**
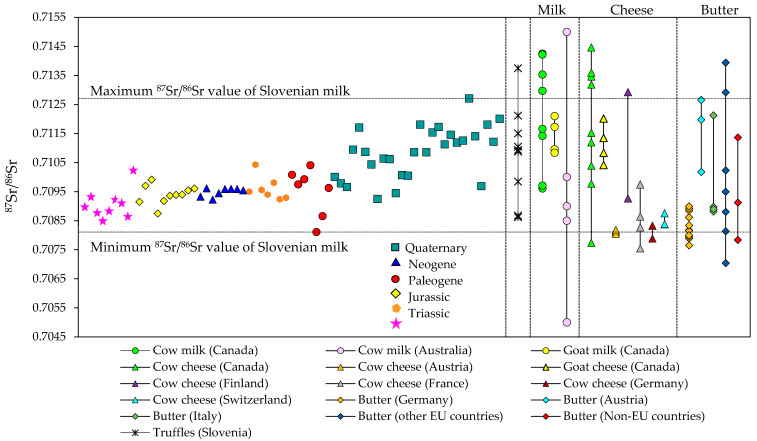
The ^87^Sr/^86^Sr isotope ratios in various dairy products from different countries, as reported in the literature. Horizontal dashed lines define the limits of the ^87^Sr/^86^Sr values measured in Slovenian milk of different geological regional origins, as indicated. References used for various dairy products worldwide: butter [24], cheese [27,30], and milk [29,30]. Dots on the vertical lines refer to the results obtained from the literature, whereas lines indicate the span of values. The ^87^Sr/^86^Sr isotope ratios in Slovenian truffles are also presented [39].

**Figure 4 foods-10-01729-f004:**
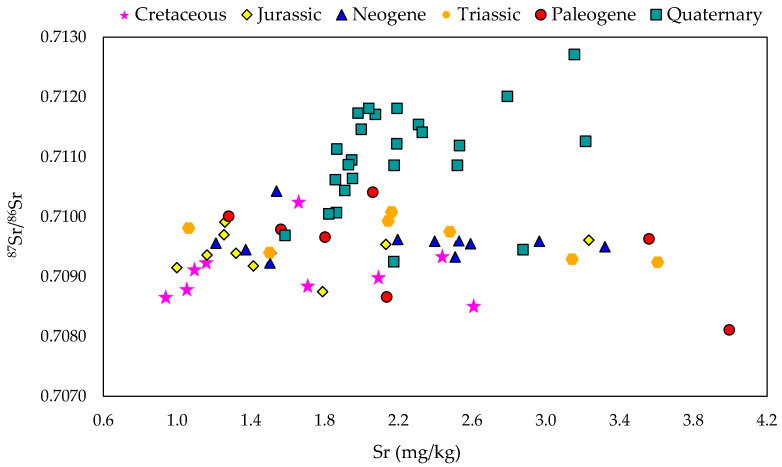
^87^Sr/^86^Sr ratios versus Sr concentrations in milk from different geological regions.

**Figure 5 foods-10-01729-f005:**
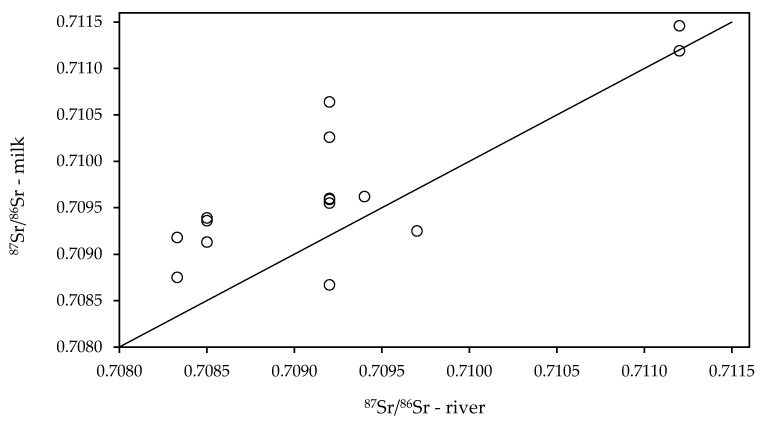
Relationship between ^87^Sr/^86^Sr ratios in rivers and milk. The line represents the 1:1 ratio indicating overlapping of the data.

**Figure 6 foods-10-01729-f006:**
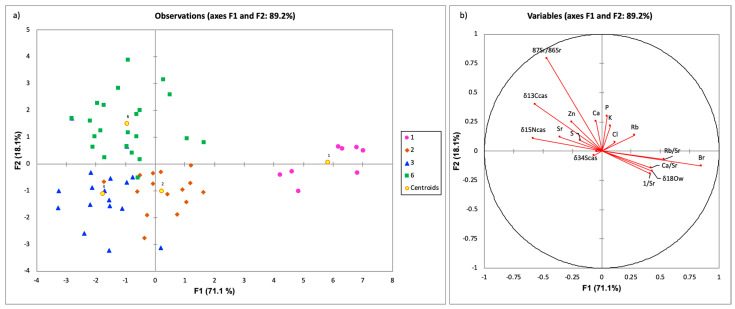
Discriminant function score plot (**a**) and a discriminant loadings plot (**b**) for milk samples collected in 2014 on four different geological origins (1—Cretaceous: Carbonate Rocks and Flysch, *n* = 8; 2—Jurassic-Triassic: Carbonate Rocks, *n* = 15; 3—Neogene: Carbonate Rocks, Paleogene: Deposits, *n* = 17; 6—Quaternary: Deposits, *n* = 24).

**Figure 7 foods-10-01729-f007:**
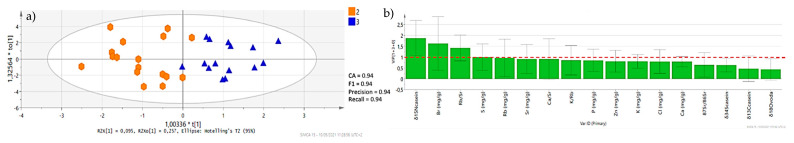
OPLS-DA score plots and VIP values in the pairwise comparison between three different geological regions derived from all isotopic and elemental composition data of milk samples. (**a**) The ellipse on the score plot represents the 95% confidence interval. (**b**) The red-dotted line indicates criteria used to identify the variables for model development.

**Table 1 foods-10-01729-t001:** Sample ID, latitude, longitude, season and year of sampling together with the content of Sr determined by ICP-MS and ^87^Sr/^86^Sr ratios in authentic milk samples. SD values are the standard deviations between samples from different farms at the same or nearby locality.

Location ID	Latitude	Longitude	Season	Year	Sr (mg/kg)	^87^Sr/^86^Sr ± SD
C1	45.70862	13.87428	summer	2014	2.09	0.70900 ± 0.00011
C1	45.70862	13.87428	winter	2014	2.44	0.70935 ± 0.00011
C2	45.60928	13.93719	summer	2014	1.05	0.70880 ± 0.00012
C2	45.60928	13.93719	winter	2014	2.60	0.70852 ± 0.00015
C2	45.60928	13.93719	winter	2015	2.26	0.70898 ± 0.00019
C3	45.55083	14.06222	summer	2014	1.71	0.70886 ± 0.00010
C3	45.55083	14.06222	winter	2014	1.16	0.70925 ± 0.00018
C3	45.55083	14.06222	winter	2015	2.02	0.70880 ± 0.00019
C4	45.77504	14.21382	winter	2014	1.09	0.70913 ± 0.00010
C4	45.77504	14.21382	winter	2015	1.79	0.70918 ± 0.00012
C5	45.91761	14.23516	summer	2014	0.94	0.70867 ± 0.00012
C5	45.91761	14.23516	winter	2014	1.65	0.71026 ± 0.00014
J1	46.30065	13.94305	summer	2014	1.00	0.70915 ± 0.00015
J2	45.83072	14.92945	summer	2014	1.25	0.70970 ± 0.00010
J2	45.83072	14.92945	winter	2014	1.26	0.70991 ± 0.00013
J2	45.83072	14.92945	winter	2015	1.21	0.70932 ± 0.00012
J3	45.97324	14.41981	summer	2014	1.79	0.70875 ± 0.00015
J3	45.97324	14.41981	winter	2014	1.41	0.70918 ± 0.00012
J4	45.85574	15.15377	summer	2014	1.16	0.70936 ± 0.00015
J4	45.85574	15.15377	winter	2014	1.32	0.70939 ± 0.00015
J5	45.46142	15.25357	summer	2014	1.51	0.70940 ± 0.00010
J5	45.46142	15.25357	winter	2014	2.13	0.70954 ± 0.00012
J6	45.97639	14.61882	winter	2014	3.23	0.70961 ± 0.00010
N1	46.68544	15.70966	summer	2014	2.51	0.70933 ± 0.00014
N1	46.68544	15.70966	winter	2014	2.20	0.70962 ± 0.00022
N2	46.22219	14.60712	summer	2014	1.50	0.70923 ± 0.00017
N2	46.22219	14.60712	winter	2014	1.37	0.70945 ± 0.00012
N3	46.25371	15.44393	summer	2014	2.40	0.70959 ± 0.00021
N3	46.25371	15.44393	winter	2014	2.53	0.70960 ± 0.00015
N4	46.23378	15.63860	summer	2014	2.96	0.70959 ± 0.00015
N4	46.23378	15.63860	winter	2014	2.59	0.70955 ± 0.00015
T1	46.55463	15.64563	winter	2014	3.32	0.70950 ± 0.00017
T2	46.18696	13.75652	summer	2014	1.54	0.71043 ± 0.00011
T2	46.18696	13.75652	winter	2014	1.21	0.70956 ± 0.00012
T3	46.04773	14.21534	winter	2014	1.50	0.70940 ± 0.00019
T3	46.04773	14.21534	winter	2015	3.41	0.70928 ± 0.00017
T4	45.83366	14.63623	winter	2014	1.06	0.70981 ± 0.00011
T5	46.18810	15.01356	summer	2014	3.61	0.70924 ± 0.00018
T5	46.18810	15.01356	winter	2014	3.14	0.70929 ± 0.00019
P1	46.53564	15.26751	summer	2014	2.16	0.71008 ± 0.00015
P1	46.53564	15.26751	winter	2014	2.48	0.70975 ± 0.00013
P2	46.59800	15.16536	winter	2014	2.14	0.70993 ± 0.00014
P4	46.50779	15.07791	winter	2014	2.06	0.71041 ± 0.00010
P5	46.33926	14.95994	summer	2014	3.99	0.70811 ± 0.00014
P5	46.33926	14.95994	winter	2014	2.14	0.70866 ± 0.00015
P6	46.42414	15.01712	winter	2014	3.56	0.70963 ± 0.00015
Q1	46.30849	14.89704	summer	2014	1.28	0.71001 ± 0.00012
Q1	46.30849	14.89704	summer	2014	1.56	0.70979 ± 0.00011
Q1	46.30849	14.89704	winter	2014	1.80	0.70966 ± 0.00012
Q2	46.54691	14.91991	winter	2014	1.95	0.71095 ± 0.00011
Q3	46.58922	15.02460	summer	2014	2.07	0.71171 ± 0.00018
Q3	46.58922	15.02460	winter	2014	1.93	0.71087 ± 0.00014
Q4	46.28804	15.03957	winter	2014	1.91	0.71044 ± 0.00015
Q4	46.28804	15.03957	winter	2015	1.92	0.70902 ± 0.00012
Q6	46.01295	15.29799	winter	2014	2.17	0.70925 ± 0.00017
Q7	46.33665	15.42204	summer	2014	1.95	0.71064 ± 0.00015
Q7	46.33665	15.42204	winter	2014	1.86	0.71062 ± 0.00011
Q8	46.39199	15.57278	winter	2014	2.88	0.70945 ± 0.00015
Q9	45.90793	15.59578	summer	2014	1.86	0.71007 ± 0.00015
Q9	45.90793	15.59578	winter	2014	1.82	0.71005 ± 0.00015
Q10	46.42006	15.86960	summer	2014	2.52	0.71086 ± 0.00023
Q10	46.42006	15.86960	winter	2014	2.04	0.71181 ± 0.00023
Q10	46.42006	15.86960	winter	2014	2.18	0.71086 ± 0.00013
Q10	46.42006	15.86960	winter	2014	2.31	0.71154 ± 0.00021
Q10	46.42006	15.86960	winter	2015	1.83	0.71010 ± 0.00013
Q11	46.67660	15.99125	summer	2014	1.98	0.71173 ± 0.00020
Q11	46.67660	15.99125	winter	2014	1.87	0.71113 ± 0.00020
Q12	46.64402	16.04111	summer	2014	2.00	0.71146 ± 0.00021
Q12	46.64402	16.04111	winter	2014	2.53	0.71119 ± 0.00025
Q13	46.65485	16.16190	summer	2014	3.22	0.71126 ± 0.00018
Q13	46.65485	16.16190	winter	2014	3.15	0.71271 ± 0.00025
Q13	46.65485	16.16190	winter	2015	2.52	0.71233 ± 0.00019
Q14	46.51955	16.19726	summer	2014	2.33	0.71141 ± 0.00019
Q15	46.28095	15.07375	winter	2014	1.58	0.70969 ± 0.00016
Q16	46.52806	15.77623	summer	2014	2.19	0.71181 ± 0.00010
Q16	46.52806	15.77623	winter	2014	2.19	0.71122 ± 0.00019
Q17	46.80051	16.22926	winter	2014	2.79	0.71201 ± 0.00020

## Data Availability

No data available.

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
