# Peer review of "The Provenance of Slovenian Milk Using 87Sr/86Sr Isotope Ratios"

_foods, 2021, doi:10.3390/foods10081729_

Round 1

Reviewer 1 Report

Overall this is important work containing Sr isotope data for the region. However, there are several issues that need to be addressed before publication:

The Sr isotope data are of lower quality (precision) than expected for Nu-Plasma MC-ICP-MS. The reported average value and reproducibility for the standard (NBS 987) is an order of magnitude worse than expected. For example, typically NBS 987 for MC-ICP-MS instruments will be 0.71024 +/-0.00003 (2-sigma error) – in contrast they report 0.71034 +/-0.00026, which is almost 10x worse error. Also not clear if this 0.00026 is 1 or 2 sigma error? Same for the data – the error is also an order of magnitude worse than expected. This is strange and not clear what is the reason? Did you use 86/88 fractionation correction during the analysis or only standard-sample-standard bracketing?  

The above issue of precision affects some of the interpretations/conclusions. For example, looking at the seasonal Sr isotope data in some cases there is difference between summer and winter. However, given the much higher error than typical then the data will be mostly within error. Therefore, the conclusion that there is no difference between summer and winter milk is not quite true (just a result of inferior Sr isotope data)..

Lines 286-288 – this is not true. The trend (if any) is actually opposite: higher 87/86 with higher Sr concentrations for the Quaternary sample. By the way, there are labels for only 3 of the symbols on fig.4. Either label all or remove the 3 labels and state in the captions that the symbols are the same as fig.3.

Lines 406-409 in the conclusions: these are two contradicting sentences. The first states that “Sr isotopes in the milk are “mixed isotope ratio of strontium from the feed and drinking water” then the next sentence states that “Sr is from the drinking water”!!

At the end, although the manuscript is overall understandable the English will need improvement.

Author Response

See attachement.

Reviewer 2 Report

See attached pdf file

Author Response

See attachement.

Round 2

Reviewer 1 Report

the paper is OK in its current form.